# Advancing Dermatomycosis Diagnosis: Evaluating a Microarray-Based Platform for Rapid and Accurate Fungal Detection—A Pilot Study

**DOI:** 10.3390/jof11030234

**Published:** 2025-03-19

**Authors:** Vittorio Ivagnes, Elena De Carolis, Carlotta Magrì, Riccardo Torelli, Brunella Posteraro, Maurizio Sanguinetti

**Affiliations:** 1Dipartimento di Scienze Biotecnologiche di Base, Cliniche Intensivologiche e Perioperatorie, Università Cattolica del Sacro Cuore, Largo A. Gemelli 8, 00168 Rome, Italy; vittorio.ivagnes@unicatt.it (V.I.); carlotta.magri@unicatt.it (C.M.); maurizio.sanguinetti@unicatt.it (M.S.); 2Dipartimento di Scienze di Laboratorio ed Ematologiche, Fondazione Policlinico Universitario A. Gemelli IRCCS, Largo A. Gemelli 8, 00168 Rome, Italy; riccardo.torelli@policlinicogemelli.it; 3Unità Operativa “Medicina di Precisione in Microbiologia Clinica”, Direzione Scientifica, Fondazione Policlinico Universitario A. Gemelli IRCCS, Largo A. Gemelli 8, 00168 Rome, Italy

**Keywords:** onychomycosis, dermatomycosis, EUROArray Dermatomycosis, fungal diagnostics, molecular diagnostics, microarray technology, rapid fungal identification

## Abstract

Dermatomycosis, including the most prevalent onychomycosis, significantly impacts patients’ quality of life due to its chronic nature and high recurrence rate. Conventional diagnostic methods are often limited by low sensitivity and specificity and prolonged turnaround times. This study evaluates the EUROArray Dermatomycosis Platform, a microarray-based molecular assay, for its performance in identifying fungi causing dermatomycosis. Forty reference fungal strains, covering on-panel and off-panel species, and 120 clinical samples from patients with suspected dermatomycosis were analyzed. The platform’s accuracy was compared to microscopy and/or culture as the diagnostic standard. The assay demonstrated 100% analytical sensitivity and 97.1% analytical specificity, correctly identifying 33 of 34 fungal species while misclassifying one. In clinical samples, the assay showed good sensitivity (78.6%) and high specificity (91.7%), detecting additional positive cases missed by culture, which highlights the assay’s ability to identify non-viable fungi and low fungal loads. The assay achieved a positive predictive value of 75.9% and a negative predictive value of 92.8%, reinforcing its diagnostic reliability. Despite some discordances, the assay provides rapid results and broad-spectrum fungal detection, positioning it as a valuable complement to conventional diagnostics. Future improvements, including expanding the identification panel and optimizing sample handling, could further enhance its clinical utility.

## 1. Introduction

Dermatomycosis refers to fungal infections affecting the skin and its adnexal structures, including nails and hair [1]. This broad category encompasses dermatophytosis, a specific subgroup of dermatomycosis. While dermatophytes represent the most common group of molds responsible for superficial infections involving keratinized tissues [2], yeasts (e.g., *Candida albicans*) and non-dermatophyte molds can also cause dermatomycosis, potentially affecting deeper layers of the skin [3].

Among dermatomycoses, onychomycosis is the most common clinical manifestation [4], with a global prevalence of approximately 5.5% [5,6,7,8,9,10]. Its incidence increases with age, ranging from 1.1% in adolescents to over 20% in adults older than 60 years [11]. Risk factors include trauma [12], diabetes [13], immunosuppression (e.g., chemotherapy, transplantation, and dialysis) [14,15,16], and dermatological conditions such as psoriasis [17].

The most frequently isolated pathogens in onychomycosis cases in North America and Europe include dermatophytes (60–70%), non-dermatophyte molds (20%), and yeasts (10–20%), with *Trichophyton rubrum* and *Trichophyton mentagrophytes* being the predominant species [6,18].

Conventional laboratory methods for diagnosing dermatomycosis include both conventional and fluorescence microscopy, fungal culture, histology, and molecular techniques [19]. Fluorescence microscopy, while rapid, is a method with low sensitivity and operator-dependent interpretation [20,21]. Fungal culture, considered the gold-standard method, can identify viable fungi but has prolonged turnaround times (up to 28 days) and sensitivity below 60% [19,22]. Histological staining techniques, such as PAS (periodic acid-Schiff) and Gomori methenamine silver (GMS), offer improved sensitivity but require specialized expertise [19,23,24].

Molecular diagnostics, particularly PCR-based methods, have enhanced fungal detection, providing greater sensitivity and specificity than conventional techniques [19]. However, their application is often limited by restricted identification panels, risk of contamination, and manual processing requirements [25,26].

The EUROArray Dermatomycosis Platform (Euroimmun Medizinische Labordiagnostika AG, Lübeck, Germany) is a microarray-based diagnostic tool designed to detect fungal pathogens from primary biological samples (nails, skin, hair) within 24 h. It amplifies fungal DNA, which is then hybridized onto a BIOCHIP microarray containing universal and species-specific probes. This enables the detection of 23 dermatophyte species, 3 yeast species, and 3 non-dermatophyte mold species with automated interpretation by the EUROArray Scanner.

This study aims to evaluate the diagnostic performance of the EUROArray Dermatomycosis Platform compared to standard mycological assays. Specifically, we assess its sensitivity, specificity, and ability to detect fungal pathogens missed by mycological culture, providing insights into its potential role in routine clinical diagnostics.

## 2. Materials and Methods

### 2.1. Study Samples

This study included 120 clinical samples from patients with suspected dermatomycosis and 40 reference strains, comprising 34 on-panel species and 6 off-panel species. The reference strains were obtained from the ATCC or CBS-KNAW collections and maintained in the clinical microbiology laboratory’s fungal strain collection. Before testing, strains were revitalized by culturing on Sabouraud dextrose agar (SDA) plates (Vacutest Kima S.r.l., Arzegrande, Italy) and checked for purity.

The clinical samples consisted of 78 nail samples (65.0%), 35 skin scrapings (29.1%), 3 skin swabs (2.5%), 2 hair samples (1.7%), and 2 skin biopsies (1.7%). These samples were prospectively enrolled from patients undergoing routine mycological examinations for suspected dermatomycosis at the clinical microbiology laboratory of Fondazione Policlinico Universitario A. Gemelli IRCCS (Rome, Italy).

No exclusion criteria were applied, ensuring that all the submitted samples were included. All the samples used in this study were residual aliquots collected in fully anonymized vials, with no access to personal patient data.

### 2.2. Standard Mycological Assays

Standard mycological assays included fluorescence microscopy, fungal culture, and the identification of isolates. Fungal culture was performed on SDA and incubated at 30 °C for up to 28 days. Growth was monitored periodically, and isolates were identified using MALDI-TOF mass spectrometry (AUTOF MS2600, Autobio Diagnostics, Zhengzhou, China) or Sanger sequencing of the internal transcribed spacer (ITS) region. For *Fusarium* species, the elongation factor 1-alpha (EF1α) gene was also sequenced, while β-tubulin sequencing was used for *Aspergillus* species [27,28].

Fluorescence microscopy was performed using 20% KOH and calcofluor white staining (Sigma-Aldrich, St. Louis, MO, USA). The samples were incubated for at least 2 h, then examined under 400× magnification using a fluorescence microscope for the presence of fungal elements [29].

### 2.3. EUROArray Dermatomycosis Platform Assay

DNA extraction was performed using the DNA MINI KIT (Qiagen, Hilden, Germany) according to the EUROArray Dermatomycosis kit protocol (Figure 1). The extracted DNA was amplified using the Dermatomycosis PCR mix, followed by hybridization on Dermatomycosis BIOCHIP microarray chips. Following hybridization, the chips were scanned using the EUROArray Scanner, and the results were interpreted through the EUROArray Scan software (version 1.1.15.231). The EUROArray can identify 23 dermatophyte species (belonging to *Trichophyton*, *Epidermophyton*, *Nannizzia*, and *Microsporum* genera), 3 *Candida* species (*C. albicans*, *C. guilliermondii*, and *C. parapsilosis*), 2 *Fusarium* species (*F. oxysporum* and *F. solani*), and 1 *Scopulariopsis* species (*S. brevicaulis*) within 24 h after clinical sample collection.

### 2.4. Analysis of Results

Regarding reference fungal strains, correct detection by the EUROArray Dermatomycosis Platform assay was defined as a match between the detected fungal species and that of the reference strain. Conversely, misdetection was defined as a discordance between the detected fungal species and that of the reference strain. According to these definitions, analytical sensitivity and specificity were calculated.

Regarding the clinical samples, the results obtained with the EUROArray Dermatomycosis Platform assay were compared with those from microscopy and/or culture (used as the comparator) to assess concordance. True positives were defined as the samples that tested positive by both the comparator and EUROArray. False positives were the samples that tested positive by EUROArray but negative by the comparator, whereas false negatives were those that tested positive by the comparator but negative by EUROArray. True negatives were the samples that tested negative by both methods. The diagnostic performance of the EUROArray assay was then assessed by calculating sensitivity, specificity, positive predictive value (PPV), and negative predictive value (NPV).

For the samples with discordant results between culture and the EUROArray Dermatomycosis Platform assay, a broad-range ITS2-based real-time PCR assay (MycoReal Kit Fungi, Ingenetix, Wien, Austria) was performed to resolve discordances. The reaction mixture included IPC3 Assay Mix (1 µL), DNA Reaction Mix (10 µL), and IPC-Target DNA (1 µL, diluted 1:100 in nuclease-free water), for a final volume of 20 µL. DNA amplification and detection were carried out using the BioRad CFX Opus 96 System (Biorad, Hercules, CA, USA).

## 3. Results

The reference fungal strain results are summarized in Table 1. All 34 on-panel fungal species were correctly identified by the EUROArray Dermatomycosis Platform assay, demonstrating a 100% sensitivity. Regarding the off-panel fungal species, none were detected, confirming the assay’s high specificity. Interestingly, *Trichophyton indotineae*, despite lacking a specific probe, was identified as *T. mentagrophytes*, which is consistent with its classification as *T. mentagrophytes* serotype VIII [30]. However, the EUROArray assay misidentified *F. solani* and produced a false positive result for *M. canis*, leading to an overall specificity of 97.1%.

Among the 120 clinical samples analyzed, 85 (70.8%) tested negative for on-panel species in both culture and the EUROArray Dermatomycosis Platform, while 35 samples (29.2%) tested positive in at least one of the two methods. Specifically, 14 samples (11.7%) were positive in both culture and the EUROArray Dermatomycosis Platform, 15 samples (12.5%) were positive in the EUROArray Dermatomycosis Platform but negative in culture, and 6 samples (5.0%) were positive in culture but negative in the EUROArray Dermatomycosis Platform. Among the 15 samples that were positive in the EUROArray Dermatomycosis Platform but negative in culture, 6 were positive for *C. parapsilosis*, 4 for *T. rubrum*, and 1 sample each for *Trichophyton interdigitale*, *F. solani*, and *Nannizia gypsea* (the teleomorph of *Microsporum gypseum*). One sample was positive for both *Trichophyton rubrum* and *Nannizia gypsea*, while another sample was positive for the dermatophyte probe only. Eight of the 15 culture-negative samples tested positive for fungal elements in microscopy examination. Regarding the six samples that were positive in culture but negative in the EUROArray Dermatomycosis Platform, two were positive for *T. rubrum*, two for *F. solani*, and one presented a mixed infection including *C. albicans*, *C. parapsilosis*, *C. guilliermondii*, and *F. solani*. The fungal species identified by both standard mycological and EUROArray Dermatomycosis Platform assays are summarized in Table 2.

For the six discordant samples (culture-positive and EUROArray-negative), pan-fungal real-time PCR did not yield cycle threshold values lower than those of the negative extraction control, with a minimum difference of 1.5 cycles for all the tested samples. This suggests that no detectable fungal DNA was present in the samples according to the EUROArray assay.

As shown in Table 3, after excluding 8 samples that grew off-panel fungal species, the remaining 112 samples were evaluated based on the results from the comparator (microscopy and/or culture). The resulting sensitivity, specificity, PPV, and NPV were 78.6%, 91.7%, 75.9%, and 92.8%, respectively.

## 4. Discussion

The EUROArray Dermatomycosis Platform assay successfully identified all 34 on-panel fungal species, which include the main etiological agents of dermatomycosis, covering both dermatophytes and non-dermatophyte fungi. The assay also correctly classified *T. indotineae* as *T. mentagrophytes*, consistent with its taxonomic placement within serotype VIII of *T. mentagrophytes*. Compared to microscopy and/or culture, the assay demonstrated high specificity (91.7%) and good sensitivity (78.6%), confirming its reliability in detecting fungi associated with dermatomycosis. Notably, microscopy revealed fungal elements in eight culture-negative samples, underscoring the value of integrating microscopy into the diagnostics of fungal infection. The ability of the EUROArray assay to detect fungal DNA regardless of viability suggests a potential advantage over culture, particularly in cases of low fungal loads or suboptimal sample collection.

The false positive and false negative results were examined to better assess the limitations of the assay. The false positive results (7 cases) may stem from cross-reactivity or the detection of non-viable fungal DNA, while the false negative results (6 cases) could be attributed to sampling errors, DNA extraction inefficiencies, or PCR inhibitors, as previously reported in molecular fungal diagnostics [25,26]. To minimize the risk of false positives, typically linked to contamination, and false negatives, often caused by PCR inhibitors, we implemented rigorous quality controls throughout the study. The internal process control embedded in the EUROArray assay ensured effective DNA recovery, while pan-fungal real-time PCR, using the previously mentioned MycoReal Kit Fungi, was performed on discordant samples to rule out inhibition effects. No appreciable fungal DNA was detected, suggesting that some culture-positive results might reflect organisms present below the detection threshold of the molecular assay. Further improvements in sample collection and DNA extraction protocols could enhance assay performance.

Compared to culture-based methods and other diagnostic tools, the EUROArray Dermatomycosis Platform assay offers several advantages. Its rapid turnaround time, providing results within 24 h, enables the timely initiation of antifungal therapy, which is crucial for improving patient outcomes. Additionally, its ability to simultaneously detect multiple dermatophytes, yeasts, and non-dermatophyte molds makes it a versatile diagnostic tool, particularly for mixed infections, which are a common challenge in clinical practice. Diagnostic challenges arise from elevated false negative rates in conventional mycological methods like microscopy and culture, particularly with non-dermatophyte mold infections [4], underscoring the importance of molecular diagnostics in this setting. However, the observed discordances between EUROArray and culture results highlight the importance of a combined diagnostic approach, integrating molecular methods with culture and clinical correlation to ensure accurate diagnoses. Although the platform has demonstrated strong diagnostic performance, its limited identification panel and sampling variability indicate areas for further optimization.

Molecular techniques, including the EUROArray Dermatomycosis Platform assay, offer high sensitivity and the ability to detect fungal DNA even in cases where viable organisms may not be recovered by culture. However, their capacity to detect fungal colonization rather than active infection raises the question of whether microscopy remains essential [29]. Given that fungi can be present on asymptomatic skin, nails, or in noninfectious nail dystrophies—particularly *Candida* species—microscopy plays a key role in distinguishing colonization from true infection. Direct examination allows for the visualization of fungal elements within host tissue structures, providing valuable context for the clinical interpretation of molecular findings [29]. For this reason, microscopy remains a critical component of the diagnostic workflow, complementing molecular and culture-based methods to ensure accurate diagnosis and prevent overtreatment.

The EUROArray Dermatomycosis Platform assay can be used with both DNA extracted from clinical samples and fungal isolates, raising the question of whether sensitivity and specificity remain consistent across sample types. While our study was designed to assess the performance of the assay on clinical samples, DNA extraction efficiency may vary between sample types due to differences in fungal load, sample matrix composition, and potential inhibitors present in clinical samples [25]. Fungal isolates, typically obtained from culture, provide a cleaner DNA source with higher concentrations, potentially improving detection rates. However, using direct clinical samples circumvents the need for culture, allowing for earlier diagnosis and treatment initiation [1]. These considerations highlight the need for further investigations comparing EUROArray assay performance on different sample types to determine the optimal conditions for diagnostic accuracy.

The relative cost of molecular assays compared to conventional culture is an important factor in their implementation in routine diagnostics. While culture-based methods remain more cost-effective, they generally require longer turnaround times (up to several weeks) and have lower sensitivity. In contrast, molecular assays, such as the EUROArray Dermatomycosis Platform, provide faster results (within 24 h) and higher sensitivity, but at a higher cost (estimated at approximately EUR 65 per test) due to the expense of specialized reagents and instrumentation. The cost-effectiveness of molecular diagnostics must, therefore, be evaluated in the context of clinical impact, as faster and more accurate diagnosis can lead to earlier treatment, reduced disease progression, and potentially lower overall healthcare costs. Future studies assessing the cost–benefit ratio of molecular testing in different healthcare settings would help define its optimal use in routine practice.

The EUROArray Dermatomycosis Platform assay represents a significant advancement in the diagnosis of dermatomycosis. Its ability to provide rapid and accurate results makes it a valuable asset in clinical microbiology laboratories. By streamlining the diagnostic workflow, it facilitates prompt and appropriate antifungal treatment, ultimately improving patient care. However, given the observed discordances between EUROArray and culture, the assay should be used in conjunction with complementary methods, particularly in complex cases or when dealing with off-panel fungal species. Future studies should focus on expanding its identification panel to include species such as *Alternaria alternata* and *T. indotineae*, and on refining protocols to further improve its diagnostic accuracy and utility in routine clinical practice. Finally, the relatively small sample size examined qualifies this as a pilot study, which nonetheless serves as a solid foundation for future larger-scale investigations aimed at further validating the assay’s clinical utility.

## Figures and Tables

**Figure 1 jof-11-00234-f001:**
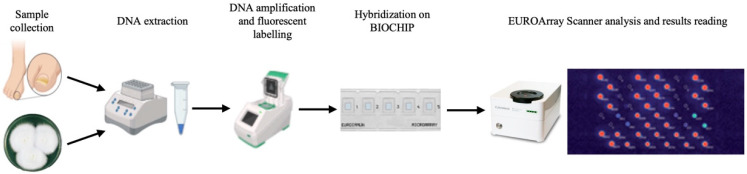
Workflow of the EUROArray Dermatomycosis Platform. The diagnostic process begins with sample collection, followed by fungal DNA extraction, amplification, and fluorescent labeling to enhance detection. The labeled DNA is then hybridized onto BIOCHIPs containing specific probes. Finally, the EUROArray Scanner analyzes the hybridized signals, enabling the rapid and accurate identification of fungal pathogens.

**Table 1 jof-11-00234-t001:** Identification of reference fungal strains using the EUROArray Dermatomycosis Platform.

Strains from Indicated Species (No. of Tested)	Results by Strains, Expressed as
	No. of Identified
On-panel species	
*Candida albicans* (2)	2
*Candida parapsilosis* (3)	3
*Fusarium oxysporum* (2)	2
*Fusarium solani* (3)	2 ^a^
*Microsporum audouinii* (1)	1
*Microsporum canis* (2)	2
*Microsporum gypseum* (2)	2
*Nannizia fulva* (1)	1
*Trichophyton erinacei* (2)	2
*Trichophyton interdigitale* (6)	6
*Trichophyton mentagrophytes* (5)	5
*Trichophyton rubrum* (3)	3
*Trichophyton tonsurans* (1)	1
*Trichophyton violaceum* (1)	1
Total species (34)	33
Off-panel species	
*Aspergillus fumigatus* (1)	0
*Fusarium delphinoides* (1)	0
*Penicillium chrysogenum* (2)	0
*Trichophyton indotineae* (2)	Both identifed as *T. mentagrophytes*
Total species (6)	0

^a^ one of the three *Fusarium solani* strains was not correctly identified.

**Table 2 jof-11-00234-t002:** Comparison of fungal species identified by the EUROArray Dermatomycosis Platform and culture.

Fungal Species	Identified by the EUROArray Platform (No. of Positive Samples)	Identified by Culture (No. of Positive Samples)
Dermatophyte (universal)	1	Not applicable
*Alternaria alternata*	0 ^a^	4
*Aspergillus niger*	0 ^a^	1
*Candida albicans*	1	2
*Candida parapsilosis*	11	5
*Candida guilliermondii*	1	1
*Fusarium solani*	2	3
*Nannizia gypsea*	2	0
*Rhizopus arrhizus*	0 ^a^	1
*Rhodotorula mucilaginosa*	0 ^a^	1
*Trichophyton interdigitale*	2	1
*Trichophyton indotinae*	1 ^b^	1
*Trichosporon mucoides*	0 ^a^	1
*Trichophyton rubrum*	9	8

For each sample, DNA was extracted and amplified by PCR, and PCR products were analyzed on a 72-spot microarray chip, which consists of 36 duplicated DNA probes (29 specific and 7 non-specific [named universal]). The results from the EUROArray Dermatomycosis Platform assay were compared with those obtained by microscopy and/or culture to calculate diagnostic performance parameters (see Table 3 for details). ^a^ off-panel fungal species, not included in the EUROArray identification panel. ^b^
*Trichophyton indotineae* was identified as *Trichophyton mentagrophytes* by the EUROArray Platform assay.

**Table 3 jof-11-00234-t003:** Diagnostic performance of the EUROArray Dermatomycosis Platform assay.

EUROArray Results Evaluated According to Those from the Comparator (Microscopy and/or Culture) ^a^	
Metric	Value
True positives (n)	22
False negatives (n)	6
False positives (n)	7
True negatives (n)	77
Results evaluated in total (n)	112
Sensitivity (%)	78.6
Specificity (%)	91.7
Positive predictive value (%)	75.9
Negative predictive value (%)	92.8

^a^ eight of EUROArray-negative samples grew isolates belonging to fungal species not included in the EUROArray Dermatomycosis Platform panel. These false negative results were excluded from the evaluation.

## Data Availability

The original contributions presented in the study are included in the article, further inquiries can be directed to the corresponding author.

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
