# Peer review of "Advancing Dermatomycosis Diagnosis: Evaluating a Microarray-Based Platform for Rapid and Accurate Fungal Detection—A Pilot Study"

_jof, 2025, doi:10.3390/jof11030234_

Round 1
Reviewer 1 Report
Comments and Suggestions for Authors
The manuscript presents the evaluation of the diagnostic performance of the EUROArray dermatomycosis platform in comparison with standard culture methods. its potential role in routine clinical diagnosis. The presentation of the work is clear. However, there are some points that the authors should address:
Material and methods
It is convenient that the positive and negative predictive values be calculated and presented
Discussion
The authors mention that a positive culture result but a negative EUROArray result can be attributed to sampling errors, inefficiencies in DNA extraction, or the presence of PCR inhibitors. However, they do not explain what actions they carried out during the study to avoid these inconveniences. What were your controls in DNA extraction, as well as your controls in PCR to guarantee absence of inhibitors?
According to Figure 1, the EUROArray can be used with both DNA extracted from clinical samples and fungal isolates. The authors should discuss whether the sensitivity and specificity are the same, regardless of the sample type. Or, discuss in terms of DNA extraction, presence of inhibitors, etc., what is most appropriate for a good diagnosis using EUROArray: clinical sample or fungal isolate?
Author Response
Comments 1:
The manuscript presents the evaluation of the diagnostic performance of the EUROArray dermatomycosis platform in comparison with standard culture methods. its potential role in routine clinical diagnosis. The presentation of the work is clear. However, there are some points that the authors should address:
Materials and Methods
It is convenient that the positive and negative predictive values are calculated and presented.
Response 1: We appreciate the reviewer’s suggestion. In response, we have calculated and included the positive predictive value (PPV) and negative predictive value (NPV) in Table 3 (newly added). These values are now explicitly referenced in the Abstract, Materials and Methods, Results, and Discussion sections (highlighted in yellow). See Table 3 and corresponding text throughout the revised manuscript.
Comments 2:
Discussion
The authors mention that a positive culture result but a negative EUROArray result can be attributed to sampling errors, inefficiencies in DNA extraction, or the presence of PCR inhibitors. However, they do not explain what actions they carried out during the study to avoid these inconveniences. What were your controls in DNA extraction, as well as your controls in PCR to guarantee absence of inhibitors?
Response 2: We appreciate this important observation. To address this point, we have clarified in the Discussion section that we implemented rigorous quality controls throughout the study. Specifically, the internal process control embedded in the EUROArray system was used to confirm successful DNA extraction and rule out PCR inhibition. Additionally, pan-fungal real-time PCR was performed on discordant samples to further exclude the presence of inhibitors. These details have been explicitly added to the revised Discussion section (highlighted in yellow).
Comments 3: According to Figure 1, the EUROArray can be used with both DNA extracted from clinical samples and fungal isolates. The authors should discuss whether the sensitivity and specificity are the same, regardless of the sample type. Or discuss in terms of DNA extraction, presence of inhibitors, etc., what is most appropriate for a good diagnosis using EUROArray: clinical sample or fungal isolate?
Response 3: We appreciate this insightful comment. In response, we have expanded the Discussion section to address the differences in sensitivity and specificity between DNA extracted from clinical samples versus fungal isolates. Specifically, fungal isolates typically yield higher DNA concentrations with fewer inhibitors, potentially improving detection rates. However, direct clinical samples enable earlier diagnosis without requiring culture, reducing turnaround time. These aspects have been explicitly discussed in the revised Discussion section (highlighted in yellow).
Reviewer 2 Report
Comments and Suggestions for Authors
I am not sure why the authors have labelled this work as an assessment of onychomycosis. A large number of the organisms detected by the EuroArray system cause other types of infection and over 30% of sites studied here were not nails. Dermatomycosis would be much more appropriate
In the discussion it would be worth seeing a very brief mention about the current role of direct microscopy. I note that the authors used fluorescence microscopy – what were the results and how did this correlate with the organisms isolated ?
We know from other studies that molecular techniques can also detect fungus that are carried at asymptomatic skin or nail sites and in nail dystrophies caused by noninfectious conditions - particularly Candida species . So do we still need microscopy ? It is worth discussing
Can you also say something about the relative costs of the test versus conventional culture ?
Author Response
Comments 1: I am not sure why the authors have labelled this work as an assessment of onychomycosis. A large number of the organisms detected by the EuroArray system cause other types of infection and over 30% of sites studied here were not nails. Dermatomycosis is much more appropriate.
Response 1: We fully agree with the reviewer’s observation. Since more than 30% of the analyzed samples originated from body sites other than nails, we have replaced “onychomycosis” with “dermatomycosis” throughout the manuscript, including in the title, Abstract, and other relevant sections. The revised title now reads: “Advancing Dermatomycosis Diagnosis: Evaluating a Microarray-Based Platform for Rapid and Accurate Fungal Detection.” Additionally, this revision required substituting the commercial name of the molecular platform with a more generic designation (“microarray-based platform”) to maintain a broader diagnostic perspective. See Title, Abstract, and relevant sections (highlighted in yellow) of the revised manuscript.
Comments 2: In the discussion it would be worth seeing a very brief mention about the current role of direct microscopy. I note that the authors used fluorescence microscopy – what were the results and how did this correlate with the organisms isolated?
Response 2: We appreciate this observation and have now included fluorescence microscopy results for the 120 clinical samples in Table 3 (newly added). These results, combined with culture, were used to classify samples as true-positive, false-positive, true-negative, and false-negative, allowing for the calculation of positive predictive value (PPV), negative predictive value (NPV), sensitivity, and specificity. Additionally, we have expanded the Discussion section to address the role of direct microscopy in fungal diagnostics, emphasizing its advantages and limitations. These modifications are now highlighted (in yellow) in the revised manuscript. See Table 3 (newly added) and Results and Discussion of the revised manuscript.
Comments 3: We know from other studies that molecular techniques can also detect fungus that are carried at asymptomatic skin or nail sites and in nail dystrophies caused by noninfectious conditions — particularly Candida species. So, do we still need microscopy? It is worth discussing.
Response 3: We appreciate this important question. To address it, we have expanded the Discussion section with a comment on the ongoing relevance of direct microscopy despite advances in molecular techniques. While molecular assays detect fungal DNA, including non-viable fungi, microscopy remains valuable for assessing fungal morphology, detecting mixed infections, and guiding initial clinical decisions. Furthermore, fluorescence microscopy results for the 120 clinical samples have been added to Table 3, as part of the classification of samples as true-positive, false-positive, true-negative, and false-negative. This addition has strengthened our performance assessment. These updates are now highlighted (in yellow) in the revised manuscript. See Table 3 (newly added) and revised Discussion section.
Comments 4: Can you also say something about the relative costs of the test versus conventional culture?
Response 4: We appreciate this suggestion. In response, we have added a brief discussion in the Discussion section, noting that while culture is more cost-effective, it has longer turnaround times and lower sensitivity for some fungi. In contrast, molecular assays like EUROArray offer faster and broader detection but at a higher cost, which may limit routine use in some settings. See revised Discussion section (highlighted in yellow).
Round 2
Reviewer 2 Report
Comments and Suggestions for Authors
The authors have answered my queries
Author Response
Comments 1: The authors have answered my queries
Response 1: Thank you!